# An Integrated Bioinformatics Analysis Repurposes an Antihelminthic Drug Niclosamide for Treating HMGA2-Overexpressing Human Colorectal Cancer

**DOI:** 10.3390/cancers11101482

**Published:** 2019-10-02

**Authors:** Stephen Wan Leung, Chia-Jung Chou, Tsui-Chin Huang, Pei-Ming Yang

**Affiliations:** 1Department of Radiation Oncology, Yuan’s General Hospital, Kaohsiung 80249, Taiwan; lwan@ms36.hinet.net; 2Graduate Institute of Cancer Biology and Drug Discovery, College of Medical Science and Technology, Taipei Medical University, Taipei 11031, Taiwan; coll78418@yahoo.com.tw (C.-J.C.); tsuichin@tmu.edu.tw (T.-C.H.); 3PhD Program for Cancer Molecular Biology and Drug Discovery, College of Medical Science and Technology, Taipei Medical University, Taipei 11031, Taiwan; 4TMU Research Center of Cancer Translational Medicine, Taipei Medical University, Taipei 11031, Taiwan; 5Cancer Center, Wan Fang Hospital, Taipei Medical University, Taipei 11696, Taiwan

**Keywords:** bioinformatics, Connectivity Map, colorectal cancer, drug repurposing, HMGA2, S100A4

## Abstract

Aberrant overexpression of high mobility group AT-hook 2 (HMGA2) is frequently found in cancers and HMGA2 has been considered an anticancer therapeutic target. In this study, a pan-cancer genomics survey based on Cancer Cell Line Encyclopedia (CCLE) and The Cancer Genome Atlas (TCGA) data indicated that HMGA2 was mainly overexpressed in gastrointestinal cancers including colorectal cancer. Intriguingly, HMGA2 overexpression had no prognostic impacts on cancer patients’ overall and disease-free survivals. In addition, HMGA2-overexpressing colorectal cancer cell lines did not display higher susceptibility to a previously identified HMGA2 inhibitor (netroposin). By microarray profiling of HMGA2-driven gene signature and subsequent Connectivity Map (CMap) database mining, we identified that S100 calcium-binding protein A4 (S100A4) may be a druggable vulnerability for HMGA2-overexpressing colorectal cancer. A repurposing S100A4 inhibitor, niclosamide, was found to reverse the HMGA2-driven gene signature both in colorectal cancer cell lines and patients’ tissues. In vitro and in vivo experiments validated that HMGA2-overexpressing colorectal cancer cells were more sensitive to niclosamide. However, inhibition of S100A4 by siRNAs and other inhibitors was not sufficient to exert effects like niclosamide. Further RNA sequencing analysis identified that niclosamide inhibited more cell-cycle-related gene expression in HMGA2-overexpressing colorectal cancer cells, which may explain its selective anticancer effect. Together, our study repurposes an anthelminthic drug niclosamide for treating HMGA2-overexpression colorectal cancer.

## 1. Introduction

High mobility group adenine–thymine (AT)-hook 2 (HMGA2) is a member of the non-histone chromosomal high mobility group (HMG) protein family and acts as an architectural factor. HMGA2 consists of three basic DNA-binding domains (AT-hooks) that bind nuclear DNA in the adenine–thymine (AT)-rich regions. HMGA2 indirectly regulates gene transcription via changing the DNA structure and allowing the assembly of protein complexes to regulate transcription [1]. HMGA2 is essential for embryonic morphogenesis and its expression is at an undetectable level in the majority of normal adult tissues [2,3]. HMGA2 is frequently overexpressed in cancers and is considered a potential therapeutic target for drug development [4,5,6,7,8,9,10,11,12,13]. However, no successful drugs have been developed to selectively kill HMGA2-overexpressing cancers.

Niclosamide is an anthelmintic drug used to treat tapeworm infestations through inhibiting glucose uptake, oxidative phosphorylation and anaerobic metabolism in the tapeworm [14]. Accumulating evidences indicate that niclosamide is a multi-targeted drug exhibiting anticancer effects. The identified targets include Wnt/β-catenin, mammalian target of rapamycin complex 1 (mTORC1), signal transducer and activator of transcription 3 (STAT3), nuclear factor kappa-light-chain-enhancer of activated B cells (NF-κB), and Notch signaling pathways [15,16]. Niclosamide has been shown to elicit anticancer activity against adenomatous polyposis coli (APC)-mutated colorectal cancer through downregulating the Wnt signaling pathway [17]. In addition, niclosamide suppresses S100 calcium-binding protein A4 (S100A4)-mediated colorectal metastasis, which also acts via intervening in the Wnt signaling pathway [18]. Recently, a phase II clinical trial has already demonstrated the safety and efficiency of niclosamide for treating metastatic colorectal cancers [19].

In this study based on the integration of bioinformatics analyses and in vitro/in vivo experimental validation, we found that HMGA2 is not a suitable prognostic biomarker and therapeutic target in colorectal cancer. However, HMGA2 overexpression in colorectal cancer cells leads to an increased sensitivity to an anthelminthic drug niclosamide. Our study provides a novel therapeutic strategy for HMGA2-overexpressing colorectal cancer.

## 2. Results

### 2.1. Pan-Cancer Analyses of HMGA2 Gene Expression

To gain a global insight into the role of HMGA2 in cancers, gene expression profiles of HMGA2 in various types of normal and cancerous cell lines and tissues were analyzed using the BioGPS (http://biogps.org) [20,21,22] and Gene Expression Profiling Interactive Analysis (GEPIA; http://gepia. cancer-pku.cn) databases [23,24]. Two datasets, GeneAtlas U133A, gcrma [25] and Cancer Cell Line Encyclopedia (CCLE) [26] with the probe 208025_s_at and 638_at, respectively were used in BioGPS analysis. The dataset GeneAtlas U133A, gcrma provides a tissue-specific pattern of mRNA expression [25]. As shown in Figure 1A, HMGA2 is highly expressed in bronchial epithelial cells, colorectal adenocarcinoma, and smooth muscle. Low HMGA2 expression is observed in other normal cells and tissues including colon. The Cancer Cell Line Encyclopedia (CCLE) dataset consists of the gene expression profiles from 917 human cancer cell line spanning 36 cancer types [26]. High expression of HMGA2 mRNA is found in various types of cancers, especially those originating from the biliary tract, large intestine (colon), esophagus, pancreas, and salivary gland (Figure 1A, the embedded graph).

To confirm the above observation, we mined The Cancer Genome Atlas (TCGA) data using the GEPIA database to compare the HMGA2 gene expression in normal and cancerous tissues. As shown in Figure 1B, HMGA2 was overexpressed in colon adenocarcinoma (COAD), esophageal carcinoma (ESCA), head and neck squamous cell carcinoma (HNSC), lung squamous cell carcinoma (LUSC), ovarian serous cystadenocarcinoma (OV), rectum adenocarcinoma (READ), skin cutaneous melanoma (SKCM), stomach adenocarcinoma (STAD), thyroid carcinoma (THCA), uterine corpus endometrial carcinoma (UCEC), and uterine carcinosarcoma (UCS). Taken together with the results of BioGPS and TCGA analyses, HMGA2 was commonly overexpressed in gastrointestinal cancers such as esophagus, stomach, colon and rectum.

### 2.2. HMGA2 Is Not a Suitable Prognosis Marker and Therapeutic Target

To investigate whether HMGA2 could serve as a prognosis marker, Kaplan–Meier survival analysis was performed. Because HMGA2 was commonly overexpressed in gastrointestinal cancers (Figure 1), we focused on esophagus, stomach, colon, and rectum cancers. Surprisingly, HMGA2 overexpression could not predict the overall and disease-free survivals of patients with esophagus (ESCA), stomach (STAD), and colorectal (COAD + READ) cancers (Figure 2). To investigate whether HMGA2 was a potential therapeutic target, we established a HMGA2-overexpressing DLD-1 colorectal cancer cell line. The overexpression of HMGA2 mRNA and protein was confirmed by real-time quantitative polymerase chain reaction (qPCR) and Western blotting (Figure 3A). The cell growth rates of HMGA2-overexpressing (DLD-1-HMGA2) and vector-overexpressing (DLD-1-Vector) cells were similar (Figure 3B). However, DLD-1-HMGA2 cells exhibited higher tumorigenicity in mice (Figure 3C), confirming the oncogenic effect of HMGA2.

HMGA2 is an AT-specific minor groove-binding protein, therefore some DNA minor groove-binding drugs, such as netropsin, are found to inhibit HMGA binding to DNA [27,28]. To investigate whether netropsin can selectively kill HMGA2-overexpressing cancer cells, both DLD-1-Vector and DLD-1-HMGA2 cells were treated with netropsin, and then cell viability was examined by a 3-(4,5-dimethylthiazol-2-yl)-2,5-diphenyl tetrazolium bromide (MTT) assay. However, the potency of netropsin toward DLD-1-Vector and DLD-1-HMGA2 cells was similar (Figure 3D). To further confirm that the effect of netropsin on colorectal cancer cell viability was not affected by HMGA2 expression, HCT116 cells with higher endogenous HMGA2 protein expression were used (Figure 3E). Consistently, similar cytotoxicity was observed in netropsin-treated DLD-1 and HCT116 cells (Figure 3F). Therefore, our results argued that HMGA2 may not a suitable prognosis biomarker and therapeutic target for colorectal cancer.

### 2.3. S100A4 Is a Potential Surrogate Therapeutic Target for HMGA2-Overexpressing Colorectal Cancer

To identify druggable vulnerability of HGMA2-overexpressing colorectal cancer, microarray analysis was performed and found that 396 gene probes were upregulated and 1450 gene probes were downregulated in DLD-1-HMGA2 cells (Appendix A). The top 10 up- and down-regulated genes are listed in Table 1. The hallmark pathways were analyzed using the Gene Set Enrichment Analysis (GSEA) software [29,30,31]. As shown in Figure 4A and Table 2, three hallmarks with false discovery rate (FDR) < 0.25 were enriched in DLD-1-HMGA2 cells. The enrichment of epithelial–mesenchymal transition (EMT) and angiogenesis was coincident with the causal role of HMGA2 in tumor metastasis [8,13]. The Venn diagram showed the four overlapped genes (*MYLK*, *BDNF*, *PCOLCE2*, and *S100A4*) among the leading-edge genes of EMT and angiogenesis, and top 10 up- and down-regulated genes (Figure 4B and the gene descriptions were shown in Table 3). Their gene expression profiles were compared by mining the BioGPS and GEPIA databases. As shown in Appendix A, *S100A4*, but not *MYLK*, *BDNF*, and *PCOLCE2*, exhibited higher expression in colorectal adenocarcinoma, but lower expression in normal colon tissues. In addition, higher *S100A4* expression was observed in various cancer types compared to *MYLK*, *BDNF*, and *PCOLCE2* (Appendix A). *S100A4* is a well-known metastasis-inducing gene in colorectal cancer and *S100A4* inhibition for therapeutic intervention is suggested [32,33]. Although *S100A4* is classified as a gene related to angiogenesis, but not EMT, in the cancer hallmark collection of Molecular Signatures Database (MSigDB) [31], it is annotated in gene ontology EMT biological process (Table 3), which can be supported by previous studies [32,34]. Therefore, we thought that *S100A4* might serve as a potential therapeutic target for HMGA2-overexpressing colorectal cancer.

To confirm the S100A4 mRNA was upregulated in HMGA2-overexpressing DLD-1 cells, qPCR was performed. There were about 1.5-fold increase of S100A4 mRNA in DLD-1-HMGA2 cells compare to DLD-1-Vector cells (Figure 4C, the left part). The corresponding induction of S100A4 protein was verified by Western blotting (Figure 4C, the embedded graph in the right part). In addition, higher extracellular level of S100A4 protein was observed in DLD-1-HMGA2 cells (Figure 4C, the right part). To investigate the prognostic impact of S100A4 on overall and disease-free survivals in colorectal cancer, Kaplan–Meier survival analysis was performed. Although S100A4 expression could not predict the overall survival, colorectal cancer patients with high S100A4 expression displayed poor disease-free survival (Figure 4D). Therefore, these results indicated that S100A4 might serve as a suitable molecular target for treating HMGA2-overexpressing colorectal cancer.

### 2.4. Connectivity Map (CMap) Analysis Identified that S100A4 Inhibition Reverses the HMGA2-Driven Gene Signature

The Connectivity Map (CMap) is a web-based database consisting of gene expression signatures from human cancer cell lines treated with small molecules or transfected with gene-encoding complementary (c)DNAs or short hairpin (sh)RNAs. By comparing the queried and existing gene signatures, the connections among genetic knockdown/overexpression or small molecules sharing similar or opposite effects can be found [35,36]. It is believed that, if the queried gene signature for a specific disease can be reversed in small-molecule-treated or gene-knockdown/overexpressing cells, the corresponding small molecules or genes (either knockdown or overexpression) might have therapeutic or suppressive effect against this disease [35,36]. Therefore, we employed CMap analysis to investigate the genetic dependency of HMGA2-overexpressing colorectal cancer. The differentially expressed genes (DEGs) with absolute Log_2_ fold change more than three (Appendix A) were queried by the next-generation CMap database, CLUE (https://clue.io/) [37]. As shown in Figure 5A, genes (knockdown or overexpression) with positive and negative scores have similar and opposite gene signatures to HMGA2 overexpression, respectively. Surprisingly, the gene signature of DLD-1-HMGA2 cells was not reversed by HMGA2-knockdown (summary score = 0.0); however, it was reversed by S100A4-knockdown (summary score = −88.19). These results suggest that HMGA2-driven gene signature can be reversed by S100A4 knockdown, but not HMGA2 knockdown. To further confirm the result of CMap prediction, a microarray expression profile from A549 cells transfected with S100A4 shRNA (GSE121628 [38]) was obtained from the Gene Expression Omnibus (GEO) database at the National Center for Biotechnology Information (NCBI) [39]. The effect of S100A4 knockdown on HMGA2-driven gene signature was analyzed using the GSEA and the enrichment plot was shown in Appendix A. We found that HMGA2-upregulated gene signature was enriched (*p* = 0.085) in sh-control cells, whereas HMGA2-downregulated gene signature was enriched (*p* = 0.03) in sh-S100A4-transfected cells, suggesting that S100A4 knockdown indeed reversed the HMGA2-driven gene signature.

Currently, several S100A4 inhibitors have been discovered through repurposing of existing drugs (Table 4). These drugs can either inhibit S100A4 expression by suppressing Wnt/β-catenin pathway such as niclosamide [18], calcimycin [40,41], and sulindac [42], or disrupting S100A4/ Myosin-IIA interaction such as phenothiazines [43,44], and NSC-95397 [45]. To investigate whether these drugs can reverse the HMGA2-driven gene signature, CMap analysis was performed. As shown in Figure 5B, only niclosamide (summary score = −87.35) showed the potency to reverse the HMGA2-driven gene signature. Specifically, the connectivity scores of niclosamide (−92.86) and S100A4 knockdown (−99.03) were similar in HT29 human colorectal cells (Figure 5B), further supporting the potency of niclosamide to inhibit S100A4 in colorectal cancer. In addition, we queried the CMap database with the S100A4-knockdown gene signature in A549 cells (GSE121628; the queried DEGs were shown in Appendix A). The summary connectivity scores of niclosamide and S100A4 knockdown were 77.95 and 96.74, respectively (Appendix A), suggesting that the queried S100A4-knockdown gene signature mimicked the signatures of niclosamide and S100A4 knockdown in the CMap database. Therefore, we speculated that S100A4 inhibition by niclosamide may exhibit therapeutic benefit for HMGA2-overexpressing colorectal cancer.

### 2.5. Connectivity Map (CMap) Analysis Identified that S100A4 Inhibition by Niclosamide is Clinically Relevant in Colorectal Cancer

To understand whether the reversion of the HMGA2-driven gene signature provides clinical benefits, the gene expression profile of a colorectal cancer patient cohort (GSE32323 [46]) was obtained from the GEO database. Next, the co-expressed genes with HMGA2 were analyzed and ranked using the GSEA software. The most similar and dissimilar genes to HMGA2 were illustrated on a heat map (Figure 6, the upper part) and the full gene list was shown in Appendix A. As expected, HMGA2 was upregulated in tumor tissues compared to the adjacent tissues. These genes were queried by the CMap database to compare the connectivity scores of HMGA2-knockdown, S100A4-knockdown, and niclosamide. As shown in Figure 6 (the lower part), the HMGA2-driven gene signature in colorectal cancer patients was reversed by S100A4-knockdown (summary score = −96.55) and niclosamide (summary score = −81.98), but not by HMGA2-knockdown (summary score = −2.63). These results were similar to those observed in HMGA2-overexpressing DLD-1 cells (Figure 5). Therefore, inhibition of S100A4 by niclosamide may have clinical benefits.

### 2.6. HMGA2-Overexpressing Colorectal Cancer Cells Are More Susceptible to Niclosamide

To validate the inhibition of S100A4 by niclosamide, DLD-1-Vector and DLD-1-HMGA2 cells were treated with various doses of niclosamide for 48 h, and then Western blot analysis were performed. As shown in Figure 7A (the upper part), niclosamide dose-dependently inhibited S100A4 protein expression. In addition, time-dependent inhibition of S100A4 protein by 1 μM niclosamide was also observed (Figure 7A, the lower part). These results were similar to the previous study that repurposes niclosamide as a S100A4 inhibitor [18]. To investigate the sensitivity of DLD-1-Vector and DLD-1-HMGA2 cells to niclosamide, cells were treated with various doses of niclosamide for 72 h, and then an MTT assay was performed. As shown in Figure 7B, DLD-1-HMGA2 cells were more sensitive to niclosamide compared to DLD-1-Vector cells. To further confirm the effect of niclosamide, HCT116 cells with higher HMGA2 and S100A4 was used (Figure 7C). Consistently, niclosamide inhibited S100A4 expression in HCT116 cells (Figure 7C), and exhibited higher cytotoxicity toward HCT116 cells compared to DLD-1 cells (Figure 7D).

To demonstrate the in vivo anticancer activity of niclosamide against HMGA2-overexpressing colorectal cancer, DLD-1-Vector and DLD-1-HMGA2 cells were transplanted simultaneously into the left and right flanks of nude mice (Figure 7E, the upper part). Transplanted DLD-1-HMGA2 cells grew faster than DLD-1-Vector cells when equal cell numbers were subcutaneously injected (Figure 3C), therefore we injected 1 × 10^6^ DLD-1-Vector and 1 × 10^5^ DLD-1-HMGA2 cells for this experiment. However, DLD-1-HMGA2 xenografts still exhibited higher tumorigenicity in mice (Figure 7E, the lower part). Treatment with niclosamide had a trend to inhibit the growth of DLD-1-Vector xenografts despite no statistical significance. In contrast, the growth of DLD-1-HMGA2 xenografts were significantly (*p* < 0.01) suppressed by niclosamide. Therefore, niclosamide indeed displayed selective in vitro and in vivo anticancer activity against HMGA2-overexpressing colorectal cancer.

### 2.7. Inhibition of S100A4 Is Not Sufficient to Selectively Kill HMGA2-Overexpressing Colorectal Cancer Cells

CMap drugs of 10 μM are usually used to treat cancer cells and generate gene expression signatures, therefore the inability of other S100A4-inhibitory drugs to reverse HMGA2-driven gene signature might be due to the higher doses required for S100A4 inhibition (Table 4). To test this possibility, the effect of sulindac was investigated. One hundred μM of sulindac has been shown to reduce S100A4 expression [42], we also found that sulindac inhibited S100A4 protein expression (Figure 8A). Surprisingly, it induced higher cytotoxicity toward DLD-1-Vector but not DLD-1-HMGA2 cells (Figure 8B). This observation implies that S100A4 inhibition cannot fully explain the effect of niclosamide. It has been proposed that niclosamide inhibits S100A4 expression by suppressing Wnt/β-catenin pathway [18]. We further examined whether Wnt/β-catenin inhibitors could exhibit selective cytotoxicity against HMGA2-overexpressing colorectal cancer cells. As shown in Figure 8C, a Wnt/β-catenin inhibitor, cardamonin, inhibited S100A4 expression, but induced HMGA2 expression. Similar to the effect of sulindac, DLD-1-HMGA2 cells were more resistant to cardamonin (Figure 8D). These results indicate that niclosamide is a better therapeutic drug for HMGA2-overexpressing colorectal cancer than other Wnt/β-catenin pathway inhibitors. In addition, inhibition of S100A4 is not sufficient to account for the effect of niclosamide.

To further confirm that role of S100A4, both DLD-1-Vector and DLD-1-HMGA2 cells were transfected with S100A4 siRNA, the reduction of S100A4 protein was ascertained by Western blot analysis (Figure 8E, the embedded figure). However, S100A4 knockdown did not exhibit higher cytotoxicity in DLD-1-HMGA2 cells as niclosamide did (Figure 8E), confirming that S100A4 downregulation was not sufficient to selectively kill HMGA2-overexpressing colorectal cancer cells. In addition, S100A4 knockdown did not further enhance the anticancer activity of niclosamide (Figure 8E). To ascertain this observation, a long-term colony formation assay was performed. Again, S100A4 knockdown did not have higher cytotoxicity against DLD-1-HMGA2 cells (Figure 8F).

### 2.8. RNA Sequencing Identifies that Niclosamide Inhibits Cell Cycle-Related Genes in HMGA2-Overexpressing Colorectal Cancer Cells

Diverse pharmacological activities of niclosamide have been continuously identified [47]. To understand the differential effect of niclosamide in HMGA2-low and HMGA2-high expressing colorectal cancer cells, RNA sequencing for niclosamide-treated DLD-1-Vector and DLD-1-HMGA2 cells were performed. There were 521 upregulated and 454 downregulated genes in niclosamide-treated DLD-1-Vector cells (Appendix A). For niclosamide-treated DLD-1-HMGA2 cells, 846 and 816 genes were upregulated and downregulated, respectively (Appendix A). The DEGs with absolute Log_2_ fold change more than two were further analyzed. The Venn diagram showed the overlapping and specific genes regulated by niclosamide in both cells (Figure 9A). These genes were analyzed for Kyoto Encyclopedia of Genes and Genomes (KEGG [48,49,50]) pathway enrichment using the WebGestalt (http://www.webgestalt.org) web tool [51,52,53]. We found that only 257 genes specifically downregulated by niclosamide in DLD-1-HMGA2 cells obtained statistically significant enrichment. As shown in Figure 9B, the most significant pathways were related to cell cycle progression (DNA replication and cell cycle) and DNA repair (homologous recombination and Fanconi anemia pathway). As a representative, the cell cycle pathway (hsa04114) mapping to niclosamide-downregulated genes in both cells was compared. As shown in Figure 9C, niclosamide upregulated two cyclin-dependent kinase (CDK) inhibitors (p15/Ink4b and p21/Cip1) and downregulated many cell cycle-related genes in DLD-1-HMGA2 cells, suggesting that it interfered with cell cycle progression. However, niclosamide only induced p21 expression but did not inhibit cell cycle-related genes in DLD-1-Vector cells (Appendix A). Therefore, we speculated that niclosamide tends to interfere with cell cycle progression and DNA repair pathways in HMGA2-overexpressing colorectal cancer, which may account for its higher cytotoxicity.

## 3. Discussion

Our results suggested that HMGA2 expression did not predict patients’ prognosis (overall and disease-free survivals) based on TCGA datasets (esophagus, stomach, colon, and rectum cancers). However, it remains controversial because negative prognostic impacts of HMGA2 overexpression in the above cancer types have been reported previously [54,55,56,57,58,59]. We hypothesized that such discrepancy may be resulted from the source of cancer patients’ data, which can be supported by a recent meta-analysis based on literatures (15 cancer types) and TCGA datasets (33 cancer types) [60]. They found that high HMGA2 expression is associated with shorter overall and disease-free survivals in cancer patients. However, consistent results are only observed in six cancer types. For clear cell renal cell carcinoma, head and neck cancer, hepatocellular carcinoma, and pancreatic ductal adenocarcinoma, patients with high HMGA2 expression have shorter overall survivals. For esophageal adenocarcinoma and ovarian carcinoma, no significant correlation was found [60].

The aim of this study was to identify therapeutic agents selectively targeting HMGA2-overexpressing colorectal cancer. HMGA2 has been considered a therapeutic target in colorectal cancer. For example, HMGA2 silencing by small interfering (si)RNA, shRNA and micro(mi)RNA (such as miR-204 and miR-4500) was shown to inhibit cell growth and induce apoptosis in human colorectal cancer cells [61,62,63,64]. However, such observations imply that HMGA2 inhibition-based therapy will be obstructed in HMGA2-overexpressing cancers. Indeed, re-expression of HMGA2 reverses the inhibitory effect of miR-204 and miR-4500 [63,64]. Our recent study has shown that heat shock protein 90 (HSP90) interacts and stabilizes HMGA2 protein expression. A HSP90 inhibitor NVP-AUY922 is able to reduce HMGA2 expression and suppress HMGA-mediated EMT in colorectal cancer cells. Thus, NVP-AUY922 can be viewed as an HMGA2 inhibitor. However, knockdown of HMGA2 by siRNA increases the sensitivity of colorectal cancer cells to NVP-AUY922, whereas HMGA2 overexpression attenuates the anticancer effect of NVP-AUY922 [65]. This study further supported the idea that the anticancer activity of HMGA2 inhibitors is limited by the endogenous expression of HMGA2 in colorectal cancer cells.

Drug repurposing is a promising strategy for the development of novel treatment modalities to fight against cancers. There are increasing repurposed drugs for colorectal cancer, including niclosamide [66]. Currently, few clinical trials of niclosamide for treating colorectal and prostate cancers are ongoing (ClinicalTrials.gov; https://clinicaltrials.gov/). Most of them are still at the recruiting stage. The only one completed clinical trial is a phase I dose-escalation study testing oral niclosamide plus standard-dose enzalutamide (a nonsteroidal antiandrogen) in metastatic castration-resistant prostate cancer [67]. However, the result is disappointing. The plasma concentration of niclosamide does not pass the threshold shown to inhibit prostate cancer growth [67]. The finding that niclosamide inhibits S100A4-driven metastasis triggers a phase II clinical trial (ClinicalTrial.gov registration number: NCT02519582) to evaluate its safety and efficacy in metastatic colorectal cancer [18,19]. This trial is expected to be completed by August 2020 and hopes to demonstrate the feasibility of oral niclosamide for treating metastatic colorectal cancer.

It has been reported that niclosamide-mediated S100A4 downregulation contributes only its inhibitory effect on cell migration and invasion, but not on cell proliferation and colony formation in HCT116 cells [18]. Similarly, we found that S100A4 silencing by siRNA did not enhance the inhibitory effect of niclosamide on cell viability. In addition, S100A4 inhibition by Wnt/β-catenin inhibitors (sulindac and cardamonin) did not exhibit higher cytotoxicity against HMGA2-overexpression DLD-1 cells. Therefore, we speculated that other mechanisms, in addition to Wnt/β-catenin and S100A4 inhibition, may involve in the selective anticancer effect of niclosamide against HMGA2-overexpressing colorectal cancer cells.

One limitation of this study is few biological replicates for microarray (two biological replicates) and RNA sequencing (two technical replicates from a single biological sample). A minimum of five biological replicates in microarray experiments is suggested to satisfy statistical power for the DEG analysis [68]. However, the most acceptable numbers for biologists are usually three. In addition, two biological replicates are considered to be acceptable for network construction only [69]. For RNA sequencing, at least six biological replicates are suggested [70]. However, a large number of published RNA sequencing studies are performed based on few biological replicates (n ≤ 2), which may not estimate the level of biological variability in gene expression [71].

## 4. Materials and Methods 

### 4.1. Bioinformatics Analysis of Public Data

Gene expression profiles of *HMGA2*, *MYLK*, *BDNF*, *PCOLCE2*, and *S100A4* in various normal and cancerous cell lines and tissues were analyzed using the BioGPS and the Gene Expression Profiling Interactive Analysis (GEPIA) databases [20,21,22,24]. For BioGPS analysis, the following datasets (and gene probes) were selected: GeneAtlas U133A, gcrma (208025_s_at for *HMGA2*, 202555_s_at for *MYLK*, 206382_s_at for *BDNF*, 219295_s_at for *PCOLCE2*, and 203186_s_at for *S100A4*) and expression data from The Cancer Cell Line Encyclopedia (CCLE) (638_at for HMGA2) [25,26]. The prognostic impact of *HMGA2* and *S100A4* in cancers was analyzed using the GEPIA database. Kaplan–Meier survival plots were generated using the GEPIA (http://gepia.cancer-pku.cn/) database [23,24], which is based on the expression data from The Cancer Genome Atlas (TCGA).

### 4.2. Microarray

Total RNA was isolated from untreated DLD-1-Vector and DLD-1-HMGA2 cells by the GENEzol TriRNA Pure Kit (Geneaid, New Taipei City, Taiwan). The mRNA profiles were analyzed using the Human OneArray Plus microarray by Phalanx Biotech (Hsinchu, Taiwan) with two biological replicates. The microarray raw data were deposited in NCBI GEO database (GSE136544). A moderated *t*-test using the limma package in R was performed to detect the differentially expressed genes (DEGs) [72]. The *p* value was adjusted for multiple testing based on the false discovery rate (FDR) according to the Benjamini-Hochberg method [73]. Appendix A listed the DEGs with |Log_2_ fold-change| > 1, *p* value < 0.01, and adjust *p* value < 0.05. For the visualization of overlap genes, Venn diagrams were generated using the VENNY 2.1 web tool (https://bioinfogp.cnb.csic.es/tools/venny/).

### 4.3. Gene Set Enrichment Analysis (GSEA) and Connectivity Map (CMap)

GSEA v2.2.2 software (Broad institute, Cambridge, MA, USA) [29,30] was used to analyze microarray data obtained from this study (GSE136544) and public resources (GSE32323 and GSE121628). The hallmark gene sets were selected for analysis [31]. Genes were ranked by running a gene set type permutation test with Log_2_ ratio ranking statistic for GSE136544 and GSE121628. For GSE32323, the most similar and dissimilar genes with HMGA2 were obtained by running a phenotype permutation test with signal-to-noise ranking statistic. Default settings were used for all other GSEA parameters. For the next-generation CMap analysis, the gene lists in Appendix A were inputted to query the CLUE database (Last accessed on September 27, 2019) [37]. Connections of HMGA2-driven gene signature to small molecules or gene knockdown/overexpression were obtained from the results and were viewed as a heat map.

### 4.4. RNA Sequencing and Kyoto Encyclopedia of Genes and Genomes (KEGG) Pathway Analysis

Total RNA was isolated from niclosamide (1 μM for 24 h)-treated and untreated DLD-1-Vector and DLD-1-HMGA2 cells by the GENEzol TriRNA Pure Kit (Geneaid, New Taipei City, Taiwan). The RNA samples were sequenced using the Illumina NovaSeq 6000 platform (San Diego, CA, USA) to generate 150 bp paired-end reads. Sequencing and data analysis were conducted by Novogene Bioinformatics Technology (Beijing, China) with two technical replicates from a single biological sample. The DEG analysis was performed using the DEGseq software [74]. Reads were normalized with trimmed mean of M-values (TMM) method and *p* values were estimated by the Poisson distribution model. The DEGs were prepared according to the criteria: |Log_2_ fold-change| > 1 and *p* value < 0.005. The full DEG list was shown in Appendix A. For the visualization of overlap genes, Venn diagrams were generated using the VENNY 2.1 web tool (https://bioinfogp.cnb.csic.es/tools/venny/). The KEGG pathway enrichment was performed using the WebGestalt web tool (Last accessed on August 28, 2019) [51,52,53]. Pathway mapping to cell cycle (hsa04114) was performed using the Advanced Pathway Painter v2.3.1 software (GSA GmbH, Rostock, Germany).

### 4.5. Materials

Roswell Park Memorial Institute (RPMI)-1640 medium, sodium pyruvate, L-glutamine, and antibiotic–antimycotic solution were purchased from Life Technologies (Carlsbad, CA, USA). Fetal bovine serum (FBS) was purchased from GIBCO (Paisley, Scotland, UK). S100A4 antibody was purchased from DAKO (Glostrup, Denmark). HMGA2, GAPDH, and β-Actin antibodies were purchased from GeneTex (Hsinchu, Taiwan). Horseradish peroxidase-labeled goat anti-rabbit and anti-mouse, and donkey anti-goat secondary antibodies were purchased from Jackson ImmunoResearch (West Grove, PA, USA). Turbo green fluorescent protein (tGFP) antibody, pCMV6-HMGA2 and pCMV6-AC-GFP plasmids were purchased from OriGene (Rockville, MD, USA). S100A4 siRNA was purchased from Santa Cruz (Island, CA, USA). PolyJet™ In Vitro DNA Transfection Reagent was purchased from SignaGen Laboratories (Gaithersburg, MD, USA). Lipofectamine RNAiMAX Reagent was purchased from Thermo Fisher Scientific (Waltham, MA, USA). Geneticin (G418) was purchased from Invivogen (San Diego, CA, USA). GENEzol TriRNA Pure Kit was purchased from Geneaid Biotech (New Taipei City, Taiwan). iScript cDNA Synthesis Kit was purchased from Bio-Rad Laboratories (Richmond, CA, USA). Netropsin, niclosamide, sulindac, dimethyl sulfoxide (DMSO), 3-(4,5-dimethylthiazol-2-yl)-2,5-diphenyl tetrazolium bromide (MTT), and crystal violet were purchased from Sigma Chemical (St. Louis, MO, USA). Cardamonin was purchased from Millipore (Bedford, MA, USA). Hybond-C Extra nitrocellulose membrane was purchased from GE Healthcare (Waukesha, WI, USA). The enhanced chemiluminescence (ECL) system was purchased from Perkin-Elmer (Seraing, Belgium). Protease and phosphatase inhibitor cocktails, X-ray film, and 2× SYBR Green PCR Master Mix were purchased from Roche (Indianapolis, IN, USA).

### 4.6. Cell Culture and Transfection

DLD-1 and HCT116 human colorectal cancer cells were cultured in RPMI-1640 medium supplemented with 10% FBS, 1 mM sodium pyruvate, 1% L-glutamine, 1% antibiotic-antimycotic solution, and incubated at 37 °C in a humidified incubator containing 5% CO_2_. Stable HMGA2-overexpressing DLD-1 (DLD-1-HMGA2) and the corresponding vector-overexpressing (DLD-1-Vector) cell lines were established by transfecting tGFP-tagged pCMV6-HMGA2 and pCMV6-AC-GFP plasmids, respectively, and then stable clones were selected with 1 mg/mL G418. For siRNA knockdown analysis, cells were transfected with Lipofectamine RNAiMAX Reagent according to the manufacturer’s instruction.

### 4.7. Determination of Cell Proliferation and Cell Viability

For the determination of cell proliferation, cells (1 × 10^5^) were plated in 60 mm dishes and cultured for 1–4 days. Cells were harvested and cell number was counted by a hematocytometer. Cell viability was examined by an MTT assay. Cells were spread on 96-well plates and exposed to drugs. After 72 h, 0.5 mg/mL of MTT was added to each well and cells were incubated for 4 h. The blue MTT formazan precipitates were dissolved in 200 μL of DMSO. The absorbance at 550 nm was measured on a multi-well plate reader. For long-term colony formation assay, 500 cells were spread in a 24-well plate and 2× serially diluted with regular medium. After 10–14 days, cells were fixed and stained with 0.5% crystal violet solution in 20% methanol.

### 4.8. Real-Time Quantitative Polymerase Chain Reaction (qPCR)

Total RNA was isolated by GENEzol TriRNA Pure Kit. The first-strand cDNA was synthesized using the iScript cDNA Synthesis Kit and then PCR was performed using the 2× SYBR Green PCR Master Mix and 200 nM of forward and reverse primers (human HMGA2: forward 5′-AGTCCCTCTAAAGCAGCTCAAAAG-3′ and reverse 5′-GCCATTTCCTAGGTCTGCCTC-3′; human S100A4: forward 5′-CTCAGCGCTTCTTCTTTC-3′ and reverse 5′-GGGTCAGCAGCTCCTTT A-3′; human β-Actin: forward 5′-GTTGCTATCCAGGCTGTGCT-3′ and reverse 5′-AGGGCATACCC CTCGTAGAT-3′). Each assay was performed on a LightCycler Nano Real-Time PCR System (Roche) in triplicate, and the fold-changes in expression were derived using the comparative cycle threshold (CT) method calculated by LightCycler Nano Software v1.1 (Roche, Indianapolis, IN, USA).

### 4.9. Western Blot Analysis and Enzyme-Linked Immunosorbent Assay (ELISA)

Cells were lysed in an ice-cold buffer (50 mM Tris-HCl (pH 7.5), 150 mM NaCl, 1 mM MgCl_2_, 2 mM EDTA, 1% NP-40, 10% glycerol, 1 mM DTT, 1× protease inhibitor cocktail and 1× phosphatase inhibitor cocktail) at 4 °C for 30 min. Cell lysates were separated on a sodium dodecyl sulfate (SDS)-polyacrylamide gel and then transferred onto the nitrocellulose membrane. The membrane was pre-hybridized in 5% skim milk/TBST (20 mM Tris-HCl, pH 7.5, 150 mM NaCl, 0.05% Tween-20) and for 1 h, and then transferred to 1% bovine serum albumin (BSA)/TBST containing a primary antibody and incubated overnight at 4 °C. After washing with the TBST buffer, the membrane was submerged in 1% BSA/TBST containing a horseradish peroxidase-conjugated secondary antibody for 1 h. The membrane was washed with TBST buffer, and then developed with an ECL system and exposed to X-ray film. The uncropped images for Western blot figures were shown in Appendix A. The densitometry readings/intensity ratio of each band were shown in Appendix A. To determine the protein level of extracellular S100A4, conditioned media from DLD-1-Vector and DLD-1-HMGA2 cells cultured for 24 h were collected and measured with ELISA commercial kits (Cloud-Clone Corp, Houston, TX, USA) according to the manufacturer’s protocol.

### 4.10. Mice Tumor Xenograft Model

DLD-1-Vector and DLD-1-HMGA2 cells were subcutaneously injected into the left and right flanks of six-week-old female nude mice (BioLASCO, Taipei, Taiwan). Mice were treated with vehicle control (5% DMSO in PBS; *n* = 5) or intraperitoneal niclosamide injection (20 mg/kg/day, 5 days/week; *n* = 5) for four consecutive weeks. Tumor length and width were measured twice per week and volume was calculated by the formula: 0.5 × length × width^2^. All animal studies were approved by the Institutional Animal Care and Use Committee at the Taipei Medical University, Taiwan (LAC-2017-0279, 16 Nov 2017).

## 5. Conclusions

In summary, our study indicated that HMGA2 is not a suitable therapeutic target although it is overexpressed in cancers. Bioinformatics analyses indicated that HMGA2-overexpressing colorectal cancers shift their dependency from HMGA2 to S100A4 that predicts a poor disease-free survival in colorectal cancer patients. In addition, niclosamide (a S100A4 inhibitor) exhibits more effective cell killing in HMGA2-overexpressing DLD-1 cells, which may provide clinical benefits. Interestingly, inhibition of S100A4 cannot fully explain the sensitizing effect of niclosamide on HMGA2-overexpressing colorectal cancer cells, suggesting that additional mechanisms are involved. The results of RNA sequencing provide an alternative explanation for the selective anticancer activity by downregulating cell cycle-related genes in HMGA2-overexpressing cells. Nevertheless, our study provides a basis for treating HMGA2-overexpressing colorectal cancer by niclosamide, and the exact mechanism of action warrants further investigation.

## Figures and Tables

**Figure 1 cancers-11-01482-f001:**
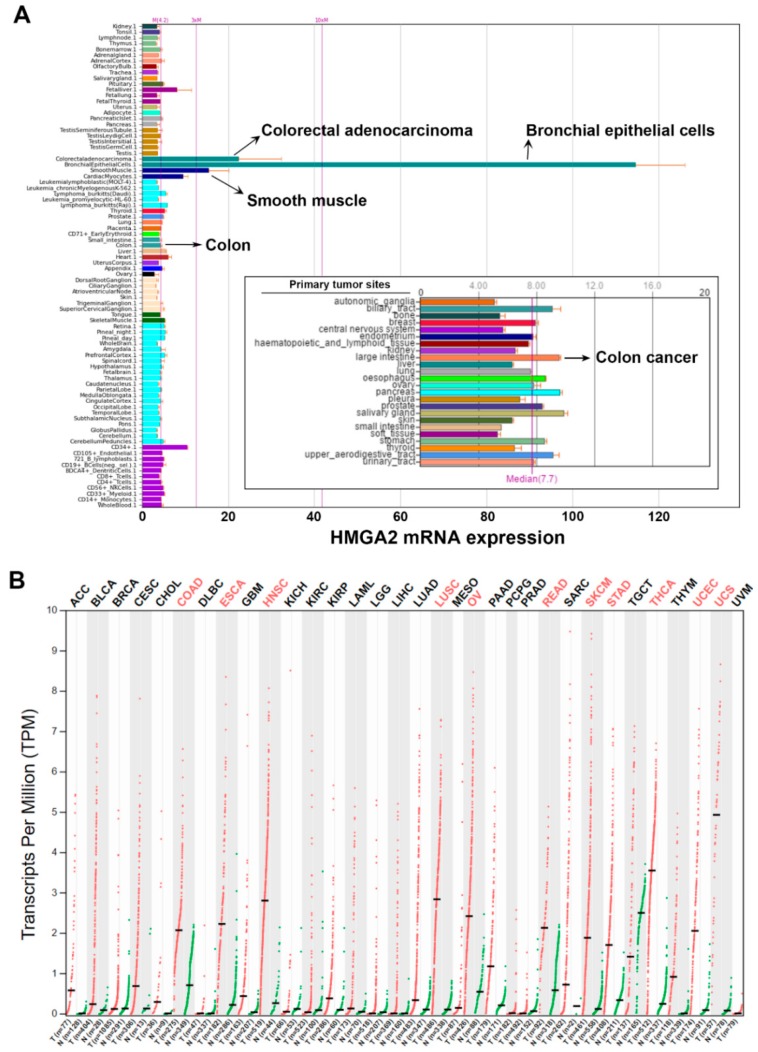
Gene expression profiles of high mobility group adenine–thymine (AT)-hook 2 (HMGA2) in normal and cancerous tissues/cells. (**A**) Gene expression profiles of HMGA2 were analyzed using the BioGPS database. Two datasets, GeneAtlas U133A, gcrma and The Cancer Cell Line Encyclopedia (CCLE) (the embedded graph), were used for analysis; (**B**) pan-cancer gene expression profiles of HMGA2 were analyzed using the Gene Expression Profiling Interactive Analysis (GEPIA) database. Significant HMGA2 overexpression in cancers are highlighted in red.

**Figure 2 cancers-11-01482-f002:**
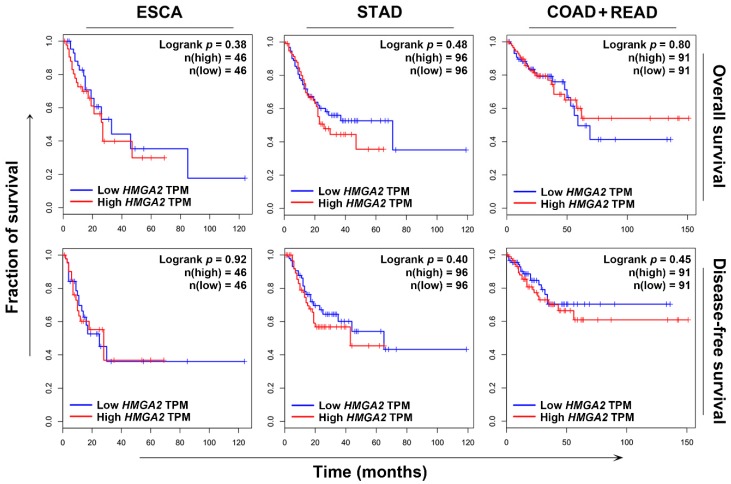
The prognostic impact of HMGA2 on overall and disease-free survival of cancer patients. The Kaplan–Meier survival plots for cancer patients with high and low HMGA2 expression were generated using the GEPIA database. TPM, transcript per million.

**Figure 3 cancers-11-01482-f003:**
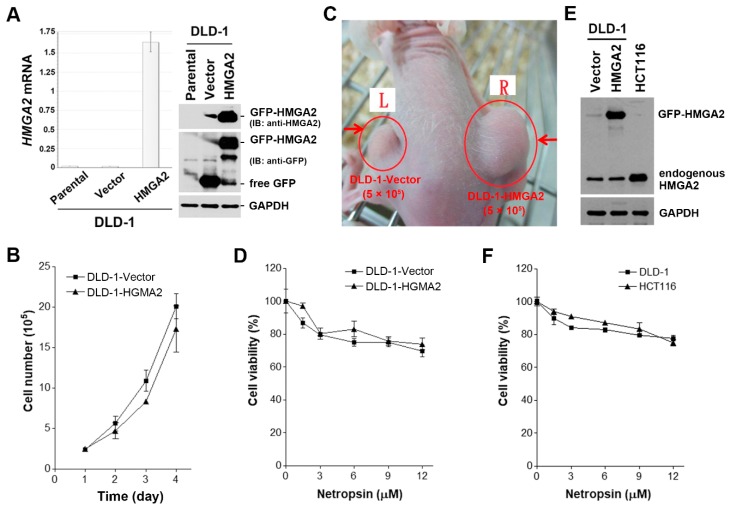
Characterization of HMGA2-overexpressing colorectal cancer cells. (**A**) The mRNA and protein expressions of HMGA2 in DLD-1-Vector and DLD-1-HMGA2 cells were analyzed by qPCR and Western blotting, respectively; (**B**) the growth rate of DLD-1-Vector and DLD-1-HMGA2 cells was measured by cell counts at 1–4 days; (**C**) the growth rate of DLD-1-Vector and DLD-1-HMGA2 cells on mice was measured using xenograft; (**D**) DLD-1-Vector and DLD-1-HMGA2 cells were treated with various doses of netropsin for 72 h. The cell viability was analyzed by a 3-(4,5-dimethylthiazol-2-yl)-2,5-diphenyl tetrazolium bromide (MTT) assay; (**E**) the protein expression of HMGA2 in DLD-1-Vector, DLD-1-HMGA2, and HCT116 cells was analyzed by Western blot analysis; (**F**) DLD-1 and HCT116 cells were treated with various doses of netropsin for 72 h. The cell viability was analyzed by an MTT assay.

**Figure 4 cancers-11-01482-f004:**
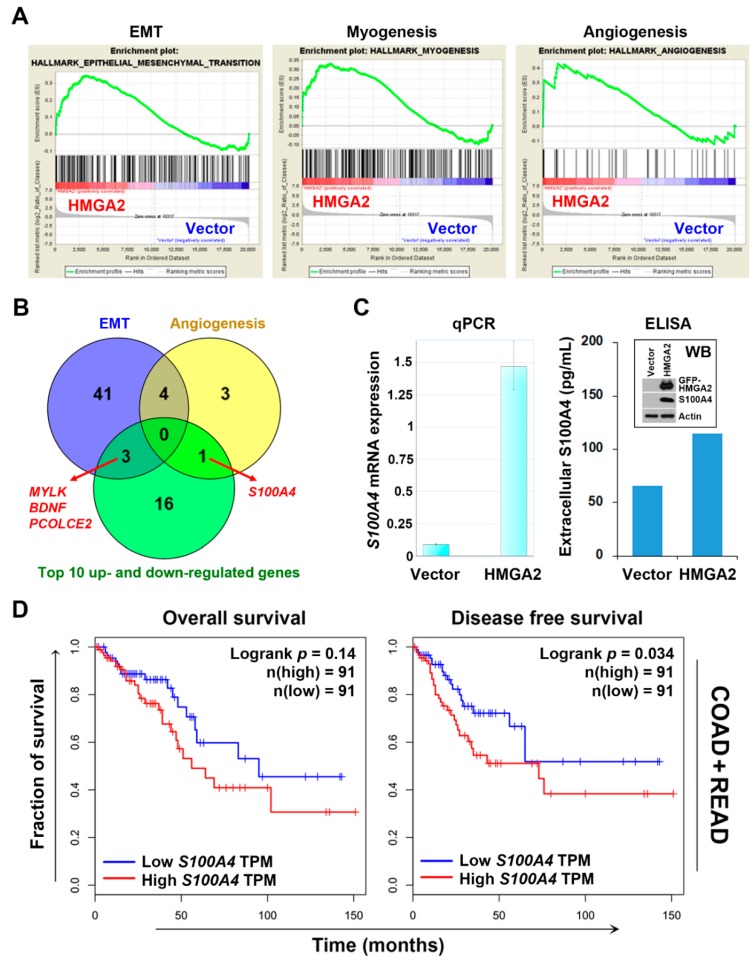
Upregulation of S100 calcium-binding protein A4 (S100A4) in HMGA2-overexpression colorectal cancer cells. (**A**) The differentially expressed genes (DEGs) in DLD-1-HMGA2 cells were analyzed by Gene Set Enrichment Analysis (GSEA) for hallmark enrichment; (**B**) a Venn diagram for the overlapped genes among epithelial–mesenchymal transition (EMT) and angiogenesis hallmarks, and top 10 up- and down-regulated DEGs in DLD-1-HMGA2 cells; (**C**) the mRNA and protein expressions of S100A4 in DLD-1-Vector and DLD-1-HMGA2 cells were analyzed by qPCR and Western blotting (WB), respectively. The level of S100A4 protein in the 24-h culture medium from DLD-1-Vector and DLD-1-HMGA2 cells was detected by enzyme-linked immunosorbent assay (ELISA); (**D**) the Kaplan–Meier survival plots for colorectal cancer patients with high and low S100A4 expression were generated using the GEPIA database. TPM, transcript per million.

**Figure 5 cancers-11-01482-f005:**
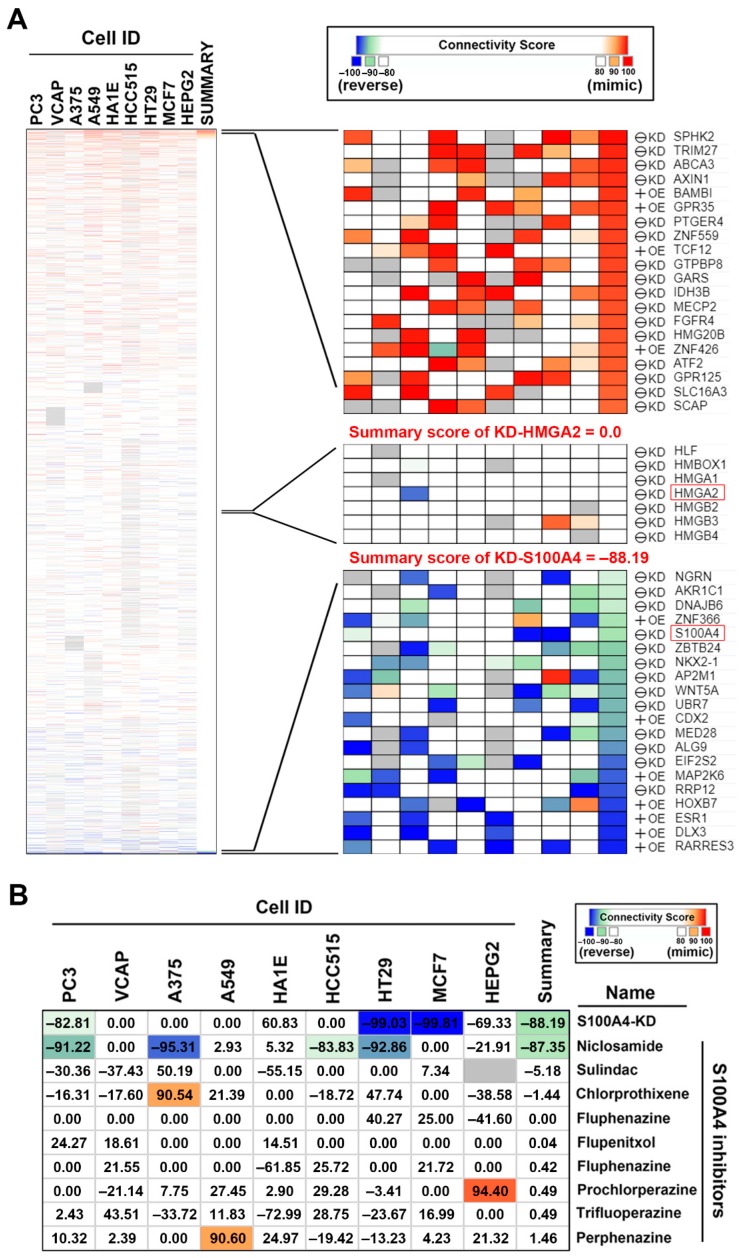
Connectivity mapping for the gene signature in DLD-1-HMGA2 cells. (**A**) Connections of HMGA2-driven gene signature with the gene knockdown/overexpression were analyzed by querying the CLUE database. Connections were viewed as a heat map ranked by the summary connectivity score; (**B**) connections of HMGA2-driven gene signature with the S100A4-inhibitory small molecules were analyzed by querying the CLUE database. Connections were viewed as a heat map with each connectivity score in individual cell line. KD, knockdown. OE, overexpression.

**Figure 6 cancers-11-01482-f006:**
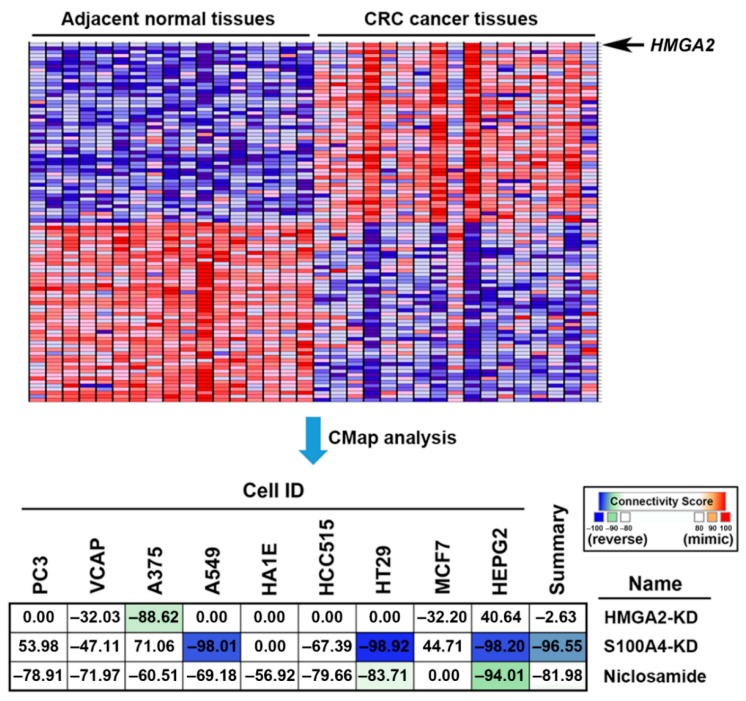
The clinical implication of S100 calcium-binding protein A4 (S100A4)-targeting niclosamide for colorectal cancer. The upper part: The most similar and dissimilar genes with HMGA2 were generated using GSEA and were visualized as a heat map. The lower part: Connections of HMGA2-driven gene signature in colorectal cancer patients with the HMGA2/S100A4 knockdown and niclosamide were analyzed by querying the CLUE database (https://clue.io/). Connections were viewed as a heat map ranked by the summary connectivity score.

**Figure 7 cancers-11-01482-f007:**
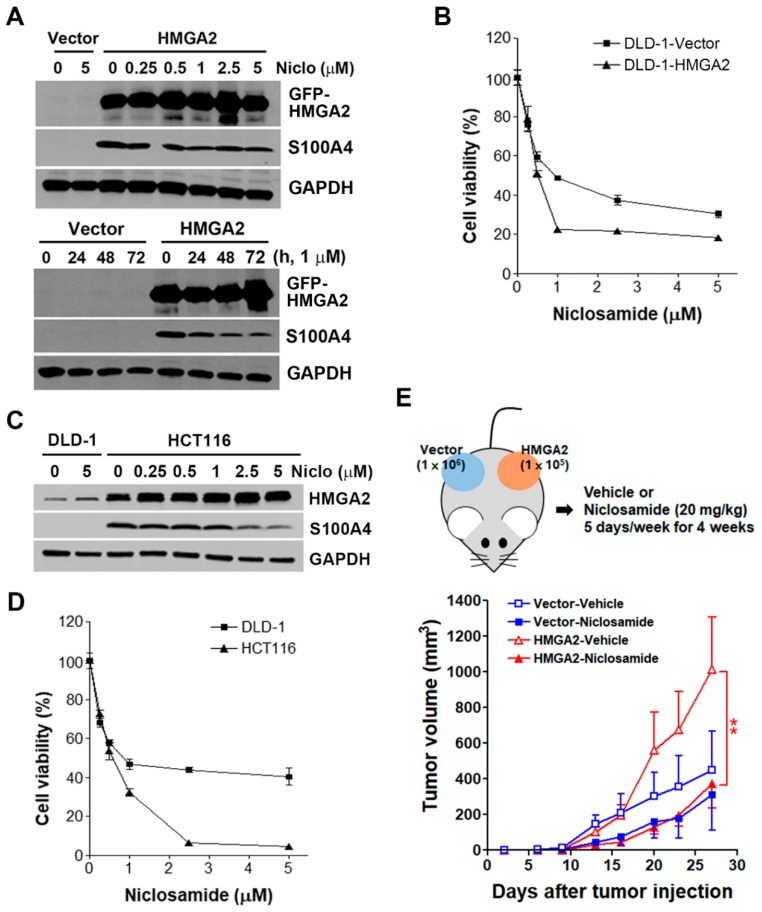
Effect of niclosamide on S100A4 expression and cancer growth in HMGA2-overexpressing colorectal cancer cells. (**A**) DLD-1-Vector and DLD-1-HMGA2 cells were treated with various doses of niclosamide for 48 h (the upper part), or 1 μM niclosamide for 1–3 days (the lower part). Whole cell lysates were prepared and subjected to Western blot analysis; (**B**) DLD-1-Vector and DLD-1-HMGA2 cells were treated with various doses of niclosamide for 72 h. The cell viability was analyzed by an MTT assay; (**C**) the protein expression of HMGA2 and S100A4 in DLD-1 and HCT116 cells was analyzed by Western blot analysis; (**D**) DLD-1 and HCT116 cells were treated with various doses of niclosamide for 72 h. The cell viability was analyzed by an MTT assay; (**E**) DLD-1-Vector (1 × 10^6^) and DLD-1-HMGA2 (1 × 10^5^) cells were subcutaneously injected into the right and left flank of nude mice. Then, mice received vehicle [5% DMSO in phosphate-buffered saline (PBS); n = 5] or niclosamide (20 mg/kg; n = 5) for four weeks. Tumor volumes were measured twice per week.

**Figure 8 cancers-11-01482-f008:**
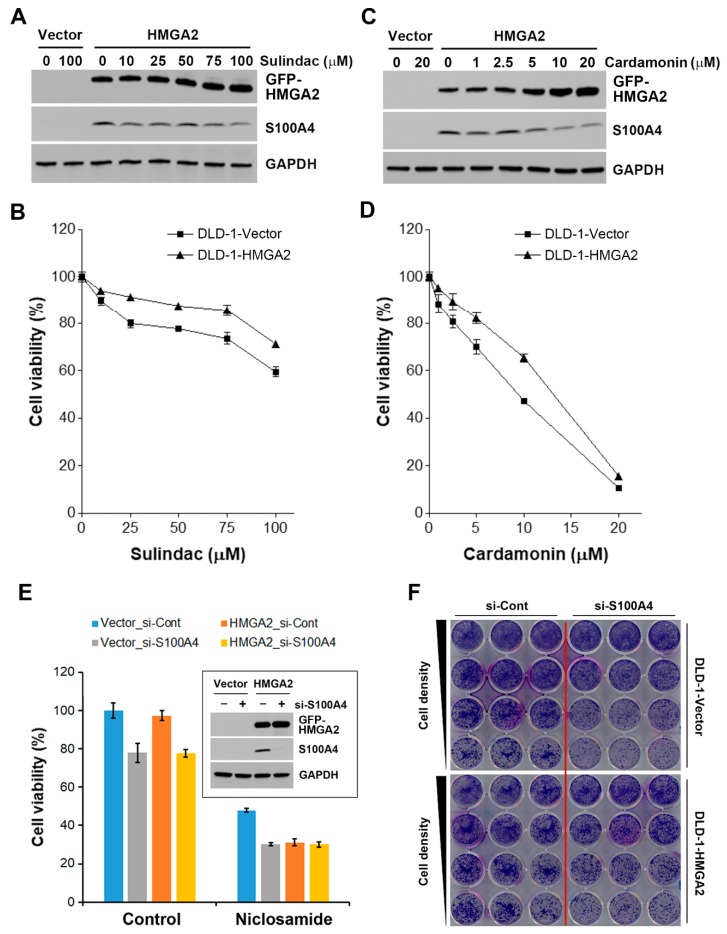
Effect of S100A4 inhibition on cell viability in HMGA2-overexpressing colorectal cancer cells. (**A**) DLD-1-Vector and DLD-1-HMGA2 cells were treated with various doses of sulindac for 48 h. Whole cell lysates were prepared and subjected to Western blot analysis; (**B**) DLD-1-Vector and DLD-1-HMGA2 cells were treated with various doses of sulindac for 72 h. The cell viability was analyzed by an MTT assay; (**C**) DLD-1-Vector and DLD-1-HMGA2 cells were treated with various doses of cardamonin for 48 h. Whole cell lysates were prepared and subjected to Western blot analysis; (**D**) DLD-1-Vector and DLD-1-HMGA2 cells were treated with various doses of cardamonin for 72 h. The cell viability was analyzed by an MTT assay; (**E**) DLD-1-Vector and DLD-1-HMGA2 cells were transfected with S100A4 siRNA for 48 h, and then treated with 1 μM niclosamide for 72 h. The cell viability was analyzed by an MTT assay; (**F**) DLD-1-Vector and DLD-1-HMGA2 cells were transfected with S100A4 siRNA for 48 h, and then colony formation assay was performed.

**Figure 9 cancers-11-01482-f009:**
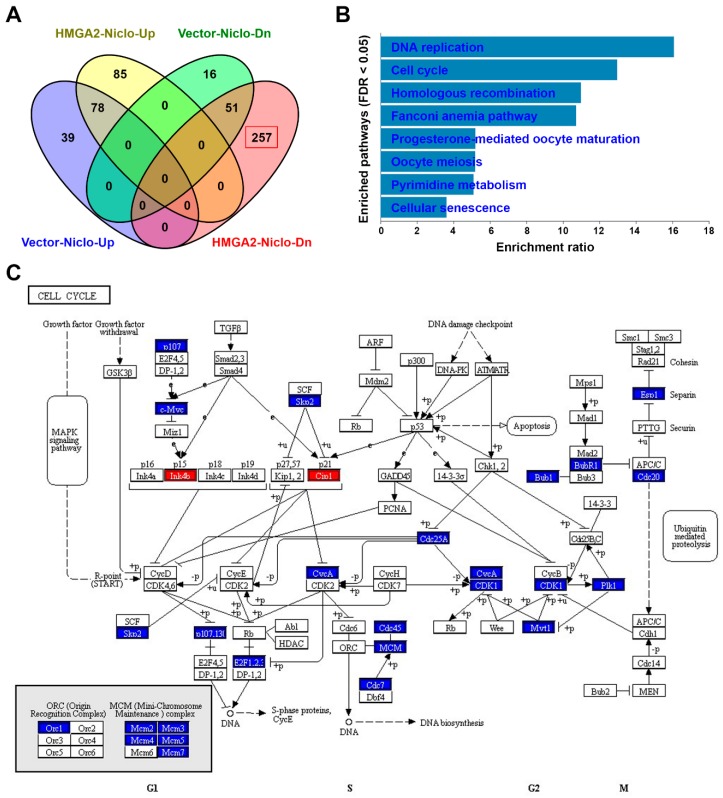
Pathway enrichment for niclosamide-treated cells. (**A**) A Venn diagram showed the numbers of overlapped genes in niclosamide-treated DLD-1-Vector and DLD-1-HMGA2 cells; (**B**) the pathway enrichment results for the downregulated genes in niclosamide-treated DLD-1-HGMA2 cells; (**C**) the mapping of cell cycle pathway for the upregulated (red) and downregulated (blue) genes in niclosamide-treated DLD-1-HGMA2 cells.

**Table 1 cancers-11-01482-t001:** Top 10 up- and down-regulated genes in high mobility group adenine–thymine (AT)-hook 2 (HMGA2)-overexpressing DLD-1 cells.

Gene Symbol	Description	Fold Change (Log_2_)
*CASQ1*	calsequestrin 1 (fast-twitch, skeletal muscle)	5.20
*QPRT*	quinolinate phosphoribosyltransferase	4.14
*S100A4*	S100 calcium binding protein A4	3.33
*MYLK*	myosin light chain kinase	3.21
*BDNF*	brain-derived neurotrophic factor	3.08
*PRKCQ*	protein kinase C, theta	2.86
*PCOLCE2*	procollagen C-endopeptidase enhancer 2	2.77
*PDE2A*	phosphodiesterase 2A	2.70
*LCN2*	lipocalin 2	2.45
*PIK3AP1*	integrin, alpha 1	2.32
*ZNF711*	zinc finger protein 711	−5.26
*PBDC1*	polysaccharide biosynthesis domain containing 1	−4.72
*NPNT*	nephronectin	−4.57
*CFTR*	cystic fibrosis transmembrane conductance regulator (ATP-binding cassette sub-family C, member 7)	−4.07
*NPTX2*	neuronal pentraxin II	−4.03
*KIT*	v-kit Hardy-Zuckerman 4 feline sarcoma viral oncogene homolog	−3.99
*SH3BGRL*	SH3 domain binding glutamic acid-rich protein like	−3.90
*METTL7A*	methyltransferase like 7A	−3.70
*CBWD1*	COBW domain containing 1	−3.62
*CDC27*	cell division cycle 27	−3.57

**Table 2 cancers-11-01482-t002:** The Gene Set Enrichment Analysis (GSEA) for hallmarks enriched in HMGA2-overexpressing DLD-1 cells.

Hallmark ^1^	Number of Genes in Pathway	Number of Pathway Genes Differentially Expressed (% of total)	NES ^2^	*p* Value	FDR ^3^ (*q* Value)
Epithelial–mesenchymal transition	197	48 (24%)	1.483	0.003	0.166
Myogenesis	198	42 (21%)	1.419	0.008	0.151
Angiogenesis	36	8 (22%)	1.373	0.083	0.154

^1^ Hallmarks with false discovery rate (FDR) < 0.25 were shown. ^2^ Normalized enrichment score. ^3^ False discovery rate.

**Table 3 cancers-11-01482-t003:** The candidate genes upregulated in HMGA2-overexpressing DLD-1 cells.

Gene	Description	Subcellular Localization	Gene Ontology (Biological Process) ^1^
*MYLK*	myosin light chain kinase	stress fiber	positive regulation of cell migration, muscle contraction
*BDNF*	brain-derived neurotrophic factor	extracellular	negative regulation of apoptotic process
*PCOLCE2*	procollagen C-endopeptidase enhancer 2	extracellular	positive regulation of peptidase activity
*S100A4*	S100 calcium binding protein A4	extracellular	epithelial to mesenchymal transition, positive regulation of I-kappaB kinase/NF-kappaB signaling

^1^ Gene information was obtained from the National Center for Biotechnology Information (NCBI) gene database (https://www.ncbi.nlm.nih.gov/gene).

**Table 4 cancers-11-01482-t004:** A list of reported S100 calcium-binding protein A4 (S100A4) inhibitors.

Drug Name	Drug Type	Mechanism for S100A4 Inhibition	Dose for S100A4 Inhibition	Dose in CMap	Reference
Niclosamide	anthelminthic agent	inhibition of the Wnt/β-catenin pathway	1 μM	20 nM~10 μM	[18]
Calcimycin (A23187)	calcium ionophore		1 μM	N.D. ^1^	[40,41]
Sulindac	nonsteroidal anti-inflammatory drug		100 μM	100 nM~10 μM	[42]
Trifluoperazine	phenothiazines	disruption of the S100A4/myosin-IIA interaction by sequestering S100A4 via small molecule-induced oligomerization	50~100 μM	100 nM~10 μM	[43,44]
Prochlorperazine				10 μM	
Perphenazine				10 μM	
Chlorprothixene				10 μM	
Flupentixol				10 μM	
Fluphenazine				100 nM~10 μM	
NSC-95397	CDC25 inhibitor		100 μM	N.D.	[45]

^1^ N.D. means not determined.

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
