# Peer review of "An Integrated Bioinformatics Analysis Repurposes an Antihelminthic Drug Niclosamide for Treating HMGA2-Overexpressing Human Colorectal Cancer"

_cancers, 2019, doi:10.3390/cancers11101482_

Round 1
Reviewer 1 Report
The paper written by Wan Leung et al. indicates that, even though HMGA2 is overexpressed in cancer, it is not a suitable therapeutic target. Moreover, the authors state that HMGA2 overexpression cannot predict the overall and disease-free survivals in patients with several types cancer. Interestingly, they identify S100A4 as the main predictor of a poor disease-free survival and a key therapeutic target in colorectal cancer overexpressing HMGA2. Thanks to these observations, Wan Leung et al. propose the use of niclosamide, a S100A4 inhibitor, for treating HMGA2-overexpressing colorectal cancer.
Overall, there were many flaws within this manuscript that make it not suitable for publication in the current form:
-Results and discussion 2.2: The authors state that "HMGA2 overexpression cannot predict the overall and disease-free survivals of patients with esophagus, stomach and colorectal cancers". Unfortunately, the major part of the literature says the contrary (i.e. Mito JK et al. 2017, Palumbo et al. 2016, Zhu et al. 2017, Nie et al. 2018, Wang et al. 2011, Pallante et al. 2016). The authors should deeply discuss this controversial statement.
-Results and discussion 2.2: Considering the results obtained treating just two HMGA2 overexpressing colon cancer cell lines only with Netropsin, the authors affirm that "HMGA2 may not be a suitable therapeutic target for colon cancer". Also this statement is against literature where HMGA2 is suggested as a therapeutic target for colon cancer (i.e. Xu et al. 2018, D'Angelo et al. 2013). The authors should deeply discuss this controversial statement.
-Results and discussion 2.2: In Figure 3 the amount of endogenous HMGA2 is different in "DLD-1 HMGA2" cells: in figure 3A it is more than the "vector" cells while in figure 3E it is the same of "vector" cells.
-Results and discussion 2.2: In Figure 3F, it should be better to not compare DLD-1 and HCT116, two different cell lines, following treatment with netropsin.
Author Response
Point 1: Results and discussion 2.2: The authors state that "HMGA2 overexpression cannot predict the overall and disease-free survivals of patients with esophagus, stomach and colorectal cancers". Unfortunately, the major part of the literature says the contrary (i.e. Mito JK et al. 2017, Palumbo et al. 2016, Zhu et al. 2017, Nie et al. 2018, Wang et al. 2011, Pallante et al. 2016). The authors should deeply discuss this controversial statement.
Response 1: We thank the Reviewer’s precious comments and suggestions. A paragraph of discussion has been added in lines 367-377. However, we did not find the suggested reference “Pallante et al. 2016”. Instead, we found a reference “Pallante et al., Front Med (Lausanne). 2015; 2: 15; High Mobility Group A Proteins as Tumor Markers”. Please let us know if we misunderstood.
Point 2: Results and discussion 2.2: Considering the results obtained treating just two HMGA2 overexpressing colon cancer cell lines only with Netropsin, the authors affirm that "HMGA2 may not be a suitable therapeutic target for colon cancer". Also this statement is against literature where HMGA2 is suggested as a therapeutic target for colon cancer (i.e. Xu et al. 2018, D'Angelo et al. 2013). The authors should deeply discuss this controversial statement.
Response 2: We thank the Reviewer’s precious comments and suggestions. A paragraph of discussion has been added in lines 378-391. However, we did not find the suggested reference “D'Angelo et al. 2013”.
Point 3: Results and discussion 2.2: In Figure 3 the amount of endogenous HMGA2 is different in "DLD-1 HMGA2" cells: in figure 3A it is more than the "vector" cells while in figure 3E it is the same of "vector" cells.
Response 3: We thank the Reviewer’s comment. The HMGA2 in Figure 3A indicated the overexpressed HMGA2-GFP detected by an anti-HMGA2 antibody. To avoid misleading the readers, we have modified Figure 3A as follows.
Point 4: Results and discussion 2.2: In Figure 3F, it should be better to not compare DLD-1 and HCT116, two different cell lines, following treatment with netropsin.
Response 4: We thank the Reviewer’s precious comments and suggestions. However, Figure 3F can support the idea that the anticancer activities of HMGA2 inhibitors (such as netropsin) or repurposed HMGA2 inhibitors (such as NVP-AUY922) are limited by the endogenous expression of HMGA2 in colorectal cancer cells. In addition, it can be compared with Figure 7D, demonstrating the ability of niclosamide to selectively inhibit the cell viability of HMGA2-overexpressing colorectal cancer cells.
Reviewer 2 Report
The authors (Leung et al.) performed various analyses to find a potential drug to treat HMGA2-overexpressing human colorectal cancer. By repurpose analysis using CMap data the antihelminthic drug niclosamide could be identified supported in vitro on transcriptional level (i.e. qPCR) and translational level (i.e. Western blot) and in vivo by mouse xenograft experiments. Analysis were based on gene expression profiling of colorectal cancer cell line overexpressing HMGA2 versus vector-overexpressing using microarrays, and the effects of niclosamide on HMGA2-overexpressing cells by RNA sequencing analyses.
This is a well written, interesting manuscript with comprehensive sound analyses underlining the conclusions. However, have some suggestions/concerns in particular for gene expression analyses and the discussion of translational context.
Major issues
It is not described how differentially expressed genes were determined for microarray analyses and RNA sequencing analyses. P-values have to be adjusted for multiple testing. Description of how RNA sequencing was performed and analyzed is missing and how many biological replicates were used. Microarray data were deposited at a public repository (Gene Expression Omnibus) from which it is obvious that only two biological replicates were used. In the manuscript should be clearly stated that only two biological replicates were used and limitations should be discussed. In 2.1 and Figure 2 it is not clear why the focus is on esophagus, stomach, and colorectal cancer and e.g. not lung cancer. Please verify metastasis-free survival in line 182 as in Figure 4D disease free survival is shown. Would suggest to separate and/or add a discussion section to include the clinical/translational context and describe therapeutic concepts (see e.g. recent publication by Fong and To, Cellular and Molecular Life Sciences (2019) 76:3383–3406) and if there are ongoing or published clinical trials or studies of niclosamide cancer treatment. Some parts from the results/discussion section can be also moved e.g. from line 252 or line 293-297Minor
Lin1 111 there should be a statement given, why HCT116 cells with higher S100A4 is analyzed here “RNA sequencing” should be used instead of “Next-generation sequencing” Number of genes in Venn diagram 16=> 6 In Figure legends there is often stated “described in Materials and Method”, but actually is not. Often is mentioned in the Results section, and can be omitted. Add normalized enrichment score (NES) and q-value in Table 2 In Table 3 S100A4 function is described as “Epithelial to mesenchymal transition” and in the hallmark in “angiogenesis” and not “EMT” please verify Line 172 though=>thought Table 4 should be rotated Line 180 KM=> Kaplan-Meier Line 195 Log2 fold change more than 3 which is different from statement in Table S2 fold-change more than ±3 please verify Line 231 As expect => as expected Line 359 , => . Line 360 NCBO=>NCBIAuthor Response
Major issues
Point 1: It is not described how differentially expressed genes were determined for microarray analyses and RNA sequencing analyses. P-values have to be adjusted for multiple testing. Description of how RNA sequencing was performed and analyzed is missing and how many biological replicates were used. Microarray data were deposited at a public repository (Gene Expression Omnibus) from which it is obvious that only two biological replicates were used. In the manuscript should be clearly stated that only two biological replicates were used and limitations should be discussed.
Response 1: We thank the Reviewer’s precious comments and suggestions. The methods for microarray and RNA sequencing have been added in lines 421-449 and 461-473. In addition, a paragraph of discussion has been added in lines 412-419 to discuss the limitation of this study.
Point 2: In 2.1 and Figure 2 it is not clear why the focus is on esophagus, stomach, and colorectal cancer and e.g. not lung cancer.
Response 2: We thank the Reviewer’s precious comment. To make it clear to the readers, the statements in lines 80-86 have been modified to describe our focus.
Point 3: Please verify metastasis-free survival in line 182 as in Figure 4D disease free survival is shown.
Response 3: We thank the Reviewer’s reminding for this error. “Metastasis-free survival” has been replaced with “disease-free survival” in lines 152 and 155.
Point 4: Would suggest to separate and/or add a discussion section to include the clinical/translational context and describe therapeutic concepts (see e.g. recent publication by Fong and To, Cellular and Molecular Life Sciences (2019) 76:3383–3406) and if there are ongoing or published clinical trials or studies of niclosamide cancer treatment. Some parts from the results/discussion section can be also moved e.g. from line 252 or line 293-297.
Response 4: We thank the Reviewer’s suggestion. A paragraph of discussion has been added in lines 392-403.
Minor issues
Point 5: Line 111, there should be a statement given, why HCT116 cells with higher S100A4 is analyzed here.
Response 5: We thank the Reviewer’s suggestion. The statement in lines 121-123 has been modified to explain the use of HCT116 cell line.
Point 6: “RNA sequencing” should be used instead of “Next-generation sequencing”.
Response 6: We thank the Reviewer’s suggestion. “Next-generation sequencing” and “NGS” have been replaced with “RNA sequencing” in this manuscript (lines 29, 343, 347, 549, 562, 563).
Point 7: Number of genes in Venn diagram 16=> 6
Response 7: We thank the Reviewer’s comments. The number of genes is correct. To avoid misleading the readers, “Top 10 genes” has been replaced with “Top 10 up- and down-regulated genes” in Figure 4B.
Point 8: In Figure legends, there is often stated “described in Materials and Method”, but actually is not. Often is mentioned in the Results section, and can be omitted.
Response 8: We apologize for the missing of methodology during manuscript preparation. The related methods have been added to “Materials and Methods” (lines 421-473). The legends in Figures 1, 2, 4, 5, 6, 8 have also been modified to remove the unnecessary statement “described in Materials and Methods”.
Point 9: Add normalized enrichment score (NES) and q-value in Table 2.
Response 9: We thank the Reviewer’s suggestion. Normalized enrichment score (NES) and q-value have been added to Table 2.
Point 10: In Table 3, S100A4 function is described as “Epithelial to mesenchymal transition” and in the hallmark in “angiogenesis” and not “EMT” please verify.
Response 10: We thank the Reviewer’s comment. Table 3 and the related statements in lines 140-146 have also been modified to explain the functional annotation for S100A4.
Point 11: Line 172, though=>thought.
Response 11: We thank the Reviewer’s reminding for this error. “though” has been replaced with “thought” in line 145.
Point 12: Table 4 should be rotated.
Response 12: We thank the Reviewer’s suggestion. Table 4 has been rotated.
Point 13: Line 180, KM=> Kaplan-Meier.
Response 13: We thank the Reviewer’s suggestion. “KM” has been replaced with “Kaplan-Meier” in line 153.
Point 14: Line 195, Log2 fold change more than 3 which is different from statement in Table S2 fold-change more than ±3 please verify.
Response 14: We thank the Reviewer’s reminding for this error. The statements in Table S2 have been corrected.
Point 15: Line 231, As expect => as expected.
Response 15: We thank the Reviewer’s reminding for this error. “As expect” has been replaced with “As expected” in line 244.
Point 16: Line 359, , => ..
Response 16: We thank the Reviewer’s reminding for this error. “,” has been replaced with “.” in line 439.
Point 17: Line 360, NCBO=>NCBI.
Response 17: We thank the Reviewer’s reminding for this error. “NCBO” has been replaced with “NCBI” in line 438.
Round 2
Reviewer 1 Report
Accept in present form
Author Response
We thank the reviewer's effort on improving our manuscript. There is no reviewer comment that we need to respond.
Reviewer 2 Report
The authors have adequately answered all raised issues and the manuscript have substantially improved and warrants the publication in Cancers.
One minor issue, I suggest the authors should address is for microarray analyses to add results from moderated t-test as e.g. used in limma package in R and adjust p-values for multiple testing e.g. based on the false discovery rate (FDR) according to the Benjamini-Hochberg method.
Author Response
Point 1: One minor issue, I suggest the authors should address is for microarray analyses to add results from moderated t-test as e.g. used in limma package in R and adjust p-values for multiple testing e.g. based on the false discovery rate (FDR) according to the Benjamini-Hochberg method.
Response 1: We thank the Reviewer’s suggestion. The original microarray result analysed by the manufacturer (Phalanx Biotech) did not perform moderated t-test and p-value adjustment. Instead, they apply Rosetta error model to conservatively estimate intensity error and improve statistical power of microarray data analysis (reference: Weng L et al., Rosetta error model for gene expression analysis.2006; 22: 1111-1121).
To address the Reviewer’s suggestion, we performed differentially expressed gene (DEG) analysis using the NCBI’s GEO2R tool to compute the values for moderated t-test and p-value adjustment. The results have been added to Table S1B. The related statement has been added in lines 422-426.